# Five millennia of *Bartonella quintana* bacteraemia

**Ba-Hoang-Anh Mai**[1,2], **Rémi Barbieri**[1], **Thomas Chenal**[3], **Dominique Castex**[4], **Richard Jonvel**[5], **Davide Tanasi**[6], **Patrice Georges-Zimmermann**[7], **Olivier Dutour**[8], **David Peressinotto**[4,9], **Coralie Demangeot**[4,9], **Michel Drancourt**[1], **Gérard Aboudharam**[1,10]*

1 Aix—Marseille Université UM63, Institut de Recherche pour le Développement IRD 198, Assistance Publique—Hôpitaux de Marseille (AP-HM), Microbes, Evolution, Phylogeny and Infection (MEPHI), Institut Hospitalo—Universitaire (IHU)—Méditerranée Infection, Marseille, France, 2 Hue University of Medicine and Pharmacy, Hue, Vietnam, 3 CNRS, UMR 6298 ArTeHiS, France, 4 UMR 5199 du CNRS, PACEA, Anthropologie des Populations Passées et Présentes, Université de Bordeaux, Pessac, France, 5 Amiens Métropole Service Archéologie Préventive, France, 6 Department of History, University of South Florida, Tampa, Florida, United States of America, 7 INRAP, UMR 5608 CNRS, TRACES, France, 8 Ecole Pratique des Hautes Etudes, Université Paris Sciences Lettres—UMR 5199 CNRS, PACEA, Université de Bordeaux, Pessac, France, 9 HADES- Bureau d'investigation archéologiques, Bordeaux, France, 10 UFR Odontologie, Aix-Marseille Université, Marseille, France

* gerard.aboudharam@wanadoo.fr

**Data Availability Statement:** All relevant data are within the paper and its Supporting Information files.

**Funding:** This study was supported by the French Government under the Investments for the Future

## Abstract

During the two World Wars, *Bartonella quintana* was responsible for trench fever and is now recognised as an agent of re-emerging infection. Many reports have indicated widespread *B. quintana* exposure since the 1990s. In order to evaluate its prevalence in ancient populations, we used real-time PCR to detect *B. quintana* DNA in 400 teeth collected from 145 individuals dating from the 1st to 19th centuries in nine archaeological sites, with the presence of negative controls. Fisher's exact test was used to compare the prevalence of *B. quintana* in civil and military populations. *B. quintana DNA* was confirmed in a total of 28/145 (19.3%) individuals, comprising 78 citizens and 67 soldiers, 20.1% and 17.9% of which were positive for *B. quintana* bacteraemia, respectively. This study analysed previous studies on these ancient samples and showed that the presence of *B. quintana* infection followed the course of time in human history; a total of 14/15 sites from five European countries had a positive prevalence. The positive rate in soldiers was higher than those of civilians, with 20% and 18.8%, respectively, in the 18th and 19th centuries, but the difference in frequency was not significant. These results confirmed the role of dental pulp in diagnosing *B. quintana* bacteraemia in ancient populations and showed the incidence of *B. quintana* in both civilians and soldiers.

## Introduction

In June 1915, a British military doctor on the western front of the First World War reported the first case of recurrent fever in a soldier who presented with a headache, dizziness, and severe pain in the lower back and leg. Additional cases with the same clinical features were

program, managed by the Agence Nationale de la Recherche (ANR) (ref: Méditerranée Infection 10-IAHU-03). This study was also supported by Région Le Sud (Provence Alpes Côte d´Azur) and European funding (FEDER BIOTK). The funders had no role in study design, data collection and analysis, decision to publish, or preparation of the manuscript.

**Competing interests:** The authors have declared that no competing interests exist.

reported among soldiers in the trenches and it began to be referred to as "trench fever" [1]. The causative agent, *Rickettsia quintana*, was isolated from a patient in Mexico City by Vinson in 1961 and was reclassified in the genus *Rochalimaea*. In 1993, *Rochalimaea* merged with *Bartonella* [2, 3]. The vector for transmission is the human body louse (*Pediculus humanus corporis*), the gastrointestinal tract of which is colonised by *B. quintana*. It is shed in lice faeces, and penetrates the human body through damaged skin, entering the bloodstream [4, 5]. The typical cycle of fever occurred at five-day intervals (hence its other name, the "five-day fever"), resulting in prolonged disability so that affected soldiers were unfit for at least two months, many of them suffering from chronic fatigue [2]. Morbidity rates were not revealed by the authorities at the time, and no deaths were reported [2, 3]. The clinical manifestations of *B. quintana* infection range from asymptomatic to a severe, life-threatening infection. These include, in addition to trench fever, lymphadenopathy, bacillary angiomatosis, and chronic bacteraemia and endocarditis [5, 6]. In the absence of any appropriate treatment, the potential for relapse is due to the existence of an intraerythrocytic phase, as erythrocytes host bacteria during their life cycle [7]. Since the 1990s, *B. quintana* has been recognised as a re-emerging agent in homeless populations as a result of unsanitary living conditions [8–11].

The presence of trench fever can be traced back to the military in Europe before the First World War, when *B. quintana* DNA sequences were found in dental pulp from Napoleonic soldiers in Vilnius (1812) [12], and in bone from soldiers in Kassel (1813–814) [13]. Indeed, dental pulp with high vascularity has been the definitive key to diagnosing bacteraemia caused by microorganisms in paleomicrobiology [14]. Molecular biology has confirmed *B. quintana* bacteraemia occurred between 200 and 4,000 years ago, through the identification of *B. quintana* in dental pulp [15]. There has been no comprehensive study of the prevalence of the pathogen over the past two millennia in Europe. In order to perform such a study, we reviewed all paleomicrobiological data regarding *B. quintana* and combined this with the study of 400 additional specimens collected from 145 individuals over 20 centuries.

## Materials and methods

### Ancient human samples

Ancient teeth were collected from human remains in nine European burial sites by archaeologists and we were granted permission by the scientific managers of these sites to conduct microbiology studies. These samples are not restricted by any regulations in force and only require the authorisation of the scientific managers of these sites (all of whom have been acknowledged as co-authors) (S1 Text).

The specimen numbers used in this study are:

Besançon–France (1st– 4th): Number from 1 to 29.

Catacombs of St. Lucia–Italy (3rd - 6th): Number from 1 to 29.

San Basilio–Italy (4th– 6th): Number from 1 to 8.

Dueville–Italy (6th– 7th): Number from 1 to 15.

Remiremont–France (5th– 10th): Number from 1 to 45.

Amiens–France (18th– 19th): Number from 1 to 55.

Dax–France (1792–1833): Number from 1 to 110.

Kaliningrad–Russia (1812): Number from 1 to 30.

Sevastopol- Ukraine (1853–1856): Number from 1 to 79.

No permits were required for the described study and this study complied with all relevant regulations.

Following our current protocol for selecting and handling teeth, teeth were washed with sterile water and gradually dried, and the dental pulp was extracted using rotating disk instruments [16]. Total DNA was extracted using the phenol-chloroform protocol [17].

## Molecular detection

Dental pulp was tested for *B. quintana* DNA using quantitative real-time PCR (qPCR) targeting the ITS gene using the following primers and probes: probe/`6FAM-GCG CGC GCT TGA TAA GCG TG-TAMRA`, forward/`5'-GAT GCC GGG GAA GGT TT TC-3'`, reverse/`5'-GCC TGG GAG GAC TTG AAC CT-3'` [18–20]. qPCR amplification was performed using the LightCycler® 480 Probes Master Kit according to the manufacturer's recommendations (Roche Diagnostics, Meylan, France). Each well contained 10 μL of mix, 3 μL of sterile water, 0.5 μL of probe (20 μM), 0.5 μL of each primer (50 μM), 0.5 μL UDG, and 5 μL of extracted DNA. Amplification consisted of a two-minute incubation step at 50°C and an initial five-minute denaturation at 95°C, followed by 40 cycles of denaturation at 95°C for five seconds and hybridisation at 60°C for 30 seconds. Two negative controls (control of DNA extraction + mix and sterile water + mix) were placed every five samples in each plate. A sample was considered positive when the qPCR result was positive with a cycle number (Ct) lower than 40.

## Fisher's exact test

Fisher's exact test was used to compare the prevalence of *B. quintana* infection in the civilian population and military population, and a p-value < 0.05 was considered statistically significant.

## Results

A total of 400 ancient teeth from 145 individuals collected in nine European sites with different times of burial were analysed using qPCR. All negative controls were negative, and qPCR detected *B. quintana* DNA in all sites. The results indicated that a total of 78 civilians from six sites had 17.9% positivity, and 67 soldiers from three sites had 20.1% positivity. The positive detection at the Besançon site was the oldest, with data from the 1st– 4th centuries (Table 1). Seven of the 17 archaeological sites had data from the 18th– 19th centuries (Tables 1 and 2), where the prevalence of *B. quintana* DNA was 3/16 (18.8%) among civilians and 24/120 (20%) among soldiers; the difference in the prevalence was not significant (Table 3).

**Table 1. Biomolecular results of *B. quintana* infection in ancient dental pulp.**

| Sites | Date | Positive teeth/total | Positive people/total | Number of teeth/person | Positive percentage | Population |
|---|---|---|---|---|---|---|
| **Besançon—France** | 1st–4th | 7/29 | 3/5 | 4–7 | 17.9% (14/78) | Civilian |
| **Catacombs of St. Lucia—Italy** | 3rd–6th | 1/29 | 1/28 | 1–2 | | |
| **San Basilio—Italy** | 4th–6th | 1/8 | 1/6 | 1–2 | | |
| **Dueville—Italy** | 6th–7th | 2/15 | 2/15 | 1 | | |
| **Remiremont—France** | 5th–10th | 13/45 | 4/8 | 3–8 | | |
| **Amiens—France** | 18th–19th | 4/55 | 3/16 | 3–4 | | |
| **Dax—France** | 1792–1833 | 11/110 | 4/9 | 9–14 | 20.1% (14/67) | Military |
| **Kaliningrad—Russia** | 1812 | 2/30 | 2/30 | 1 | | |
| **Sevastopol- Ukraine** | 1853–1856 | 14/79 | 8/28 | 1–5 | | |

**Table 2. *Bartonella quintana* detection in ancient specimens from previous studies.**

| Sites | Date | Specimens | Methods | Positive number of specimens/ total | Positive number of people/ total | Population | Ref. |
|---|---|---|---|---|---|---|---|
| Peyraoutes—France | 2000BCE | Dental pulp | Suicide PCR (groEL, hbpE genes) | 1/6 | 1/3 | - | [21] |
| Roaix—France | 2200BCE–2100BCE | | | 0 | 0/3 | | |
| Bondy—France | 11th–15th | | Real time PCR (ITS gene) | 3/14 | 3/5 | Civilian | [18] |
| Venice—Italy | 15th–16th | | | 5/173 | - | Civilian | [19] |
| Douai—France | 18th | | | 1/40 | - | Military | [20] |
| Vilnius—Lithuania | 1812 | | Suicide PCR (hbpE, htrA genes) | 7/72 | 7/35 | Military | [12] |
| | | Lice | PCR standard (hbpE gene) | | | | |
| Kassel -Germany | 1813–1814 | Bone | PCR standard (hbpE gene) | 3/18 | 3/18 | Military | [13] |
| Namur—Belgium | 14th | Coprolite | Metagenomics | | | | [22] |

(-): Not mentioned

## Discussion

The results reported here were authenticated by the fact that we chose ancient teeth, preferentially monoradicular teeth with a close apex, no traumatic lesions, and an absence of dental caries that helped minimise any risk of external contamination [16]. A positive control was not used in the PCR experiments because it could be a source of contamination; the negative controls remained negative [18]. To screen a total of 400 teeth, we used real-time PCR as previously described to identify *B. quintana* [18–20], and in this work, each experimental step was performed in different rooms of a new building, where these gene sequences had not previously been used. The number of teeth per individual was variable in each burial site due to availability, and we used as many teeth per individual as possible to increase the chance of detection. Indeed, the first report of using dental pulp for ancient septicaemia diagnosis indicated that the number of teeth per individual infected with *Yersinia pestis* was 1/4 teeth, 1/2 teeth, and 3/3 teeth [14]. Furthermore, two teeth from one individual reported that the first was infected with *Y. pestis* and the second had co-infection of *Y. pestis* and *B. quintana* [18]. Accordingly, in several instances, co-detection of *B. quintana* with another deadly pathogen has been reported, such as *Yersinia pestis*, *Rickettsia prowazekii*, indicating that the detection of *B. quintana* does not preclude that of other pathogens in any archaeological site [12, 18, 23].

In 1915, the history of trench fever was marked by the report of new clinical features from soldiers involved in trench warfare [2, 3]. In 2005, authentic evidence showed that *B. quintana* had caused bacteraemia in humans for more than 4,000 years, through analysis of DNA from dental pulp collected from Peyraoutes in France [21]. This pathogen was also later identified in this specimen from different burial sites, including Venice (15th–16th centuries) [19], Douai (18th century) [20], Vilnius (1812) [12] and Bondy (11th–15th centuries) where an individual had coinfection with *Y. pestis* [18]. In addition to dental pulp, *B. quintana* was detected in

**Table 3. Comparison of infected populations of 18th–19th centuries.**

| Population | Total | Positive number | Percentage | p |
|---|---|---|---|---|
| Civilians | 16 | 3 | 18.8% | > 0.05 |
| Soldiers | 120 | 24 | 20% | |

ancient bone (1813–1814) [13] and coprolites (14th) century [22]. The bacterial confirmation in ancient pulp indicates that individuals had bacteraemia before their death, but it may not have been the cause of death, supporting the fact that no deaths have been recognised as being caused by trench fever [2, 5, 24]. A survey of 930 homeless people in Marseille revealed that 5.3% were blood culture positive for *B. quintana* [8], and asymptomatic chronic bacteraemia could be maintained for 78 weeks [25]. In 2004, *B. quintana* was found in the dental pulp of a homeless patient who had had bacteraemia in the previous six months [26].

Most of the reports of *B. quintana* in dental pulp come from our laboratory, with nearly half of burial sites being located in the 18[th]–19[th] centuries; 7/14 sites were in France, 4/14 were in Italy, and the rest came from Ukraine, Russia, and Lithuania. This can be explained by the availability of samples in these periods, and the conditions in which our research team in France operate made it highly favourable to receiving these teeth from archaeological centres. The wide geographical and temporal distribution suggests that this bacterial infection was common in historic European populations. In the medical literature, the "five-day fever" and the Moldavia fever in the 19[th] century had clinical signs similar to trench fever but laboratory techniques to confirm this were lacking at that time [27].

Millions of soldiers were contaminated around the world during the two World Wars. After the end of each war, however, incidence dropped dramatically but still sporadically persisted in some countries [5, 24]. It was noted that Napoleonic soldiers buried in Vilnius, Lithuania (1812) were exposed to body lice containing *B. quintana*, and approximately 20% and 8.6% of soldiers were infected with *B. quintana* and *R. prowazekii*, respectively [12]. An investigation of ancient bones from 18 soldiers from a mass grave inhumed in the winter of 1813–14 in Kassel, Germany, revealed that 16.7% of the soldiers had the infection [13]. In our study, only 6.7% of the soldiers from the Kaliningrad site tested positive, in contrast to the higher positive rates from Dax and Sevastopol. This can be explained by the use of only one tooth per individual for the Kaliningrad samples. The catacombs in St. Lucia, which represented the ancient Christian monument of the Late Roman period [28], had the similarity of also using one tooth, with the lowest positive proportion (3.6%). During wartime, soldiers were generally crowded close together in unsanitary conditions for prolonged periods of time. Moreover, despite knowing the role that lice played in transmission and their presence in clothes, there were no effective methods of disinfection available [2, 3].

In order to ensure we were working with a uniform time period, we chose the 18[th]–19[th] centuries, which had a greater number of both populations. The incidence in civilians and soldiers was 18.8% and 20%, respectively, and there was no significant difference. This suggests that this bacterial infection was relatively common throughout human society, without any detectable difference in lice infestation and clothing hygiene, although the sample size of citizens was small (18 citizens), which could affect the comparison. The epidemiology of trench fever involves interhuman transmission via vectors, such as body louse infection, or other transmission that may involve contact with reservoirs, insects bites [29], cat bites [30] or others that have yet to be defined and need further elucidation. Humans are not the only reservoirs of *B. quintana*; it is also found in cat fleas [31] and monkey fleas [32], and some studies have indicated other animals as known reservoirs through the detection of *B. quintana* in domestic cats [33], dogs [34], rhesus macaques [35, 36], and Japanese macaques [37]. Following an experimental study on cat fleas, it was shown that this pathogen was absorbed via the gastrointestinal tract and released into faeces [38]. Another study showed that the *Pedicinus obtusus* louse was postulated as an efficient vector of transmission between rhesus macaques [36]. Homelessness was defined as a high-risk factor for *B. quintana* infection as a consequence of inadequate hygiene [8, 10, 11, 25]; however, some subpopulations have relatively high exposure rates, such as blood donors, with a rate of between 27% and 51% [9, 39, 40], and healthy people, with a

rate of between 11.2% and 25% [40–43]. There is no evidence to show that these people lived in unhygienic living conditions, such as being homeless, or that they had come into contact with body lice, suggesting the existence of some underestimated factors that cause a bacterial infection. The new findings on transmission, transmitted vectors, reservoirs, and widely infected populations contribute to the understanding of the infection from the past to the present; paleomicrobiological studies as this one helping to clarify complex host-pathogen interactions in past human populations, as previously reported [44, 45].

## Conclusions

Dental pulp is conducive to the investigation of *B. quintana* bacteraemia in ancient populations, and it is relatively easy to collect and is well protected inside the tooth. Previous studies analysed in this study showed the presence of *B. quintana* infection 4,000 years ago and between the 1st and 19th centuries. Soldiers are considered the main target of this bacterial infection in human history, but in this study, we saw no significant difference between civilians and soldiers during the 18th and19th centuries.

## Supporting information

**S1 Text. Nine European archeological sites.**
(DOCX)

## Acknowledgments

Our thanks go to Michael Decker (University of South Florida, Department of History, Tampa, Florida, USA) for his information on the samples. Thanks also go to the Pontifical Commission for Sacred Archaeology–Inspectorate for the Catacombs of Eastern Sicily for the permission to work on other samples from the Catacombs of Saint Lucia of Syracuse, Italy.

## Author Contributions

**Conceptualization:** Michel Drancourt, Gérard Aboudharam.

**Data curation:** Ba-Hoang-Anh Mai, Rémi Barbieri, Michel Drancourt, Gérard Aboudharam.

**Formal analysis:** Ba-Hoang-Anh Mai.

**Investigation:** Ba-Hoang-Anh Mai, Michel Drancourt, Gérard Aboudharam.

**Methodology:** Michel Drancourt, Gérard Aboudharam.

**Project administration:** Michel Drancourt, Gérard Aboudharam.

**Resources:** Thomas Chenal, Dominique Castex, Richard Jonvel, Davide Tanasi, Patrice Georges-Zimmermann, Olivier Dutour, David Peressinotto, Coralie Demangeot, Michel Drancourt, Gérard Aboudharam.

**Supervision:** Michel Drancourt, Gérard Aboudharam.

**Validation:** Michel Drancourt, Gérard Aboudharam.

**Writing – original draft:** Ba-Hoang-Anh Mai, Rémi Barbieri, Thomas Chenal, Dominique Castex, Richard Jonvel, Davide Tanasi, Patrice Georges-Zimmermann, Olivier Dutour, Michel Drancourt, Gérard Aboudharam.

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
