## [Decision Letter · Decision Letter 0]

23 Apr 2020

PONE-D-20-07460

Five millennia of Bartonella quintana bacteremia.

PLOS ONE

Dear Dr. Aboudharam,

Thank you for submitting your manuscript to PLOS ONE. After careful consideration, we feel that it has merit but does not fully meet PLOS ONE’s publication criteria as it currently stands. Therefore, we invite you to submit a revised version of the manuscript that addresses the points raised during the review process.

We would appreciate receiving your revised manuscript by Jun 07 2020 11:59PM. To enhance the reproducibility of your results, we recommend that if applicable you deposit your laboratory protocols in protocols.io, where a protocol can be assigned its own identifier (DOI) such that it can be cited independently in the future. For instructions see: http://journals.plos.org/plosone/s/submission-guidelines#loc-laboratory-protocols

We look forward to receiving your revised manuscript.

Kind regards,

David Caramelli, Ph.D

Academic Editor

PLOS ONE

2. Thank you for including the following ethics statement on the submission details page:

'all the teeth studied, we were provided by their scientific managers having the full

liberty to conducted microbiology studies. This kind of sample is not restricted by any

regulation in force and only requires the authorization of the scientific manager (all

granted as co-authors)'

Please also include this information in the ethics statement in the Methods section of your manuscript.

3. Please include your tables as part of your main manuscript and remove the individual files. Please note that supplementary tables (should remain/ be uploaded) as separate "supporting information" files

"This work was supported by the French Government under the Investissements d’avenir (Investments for the Future) program managed by the Agence Nationale de la Recherche (ANR, fr: National Agency for Research), [reference: Méditerranée Infection 10-IAHU-03]. This work was supported by Région Le Sud (Provence Alpes Côte d’Azur) and European funding [FEDER BIOTK]."

6. We note you have included a table to which you do not refer in the text of your manuscript. Please ensure that you refer to Table 1-3 in your text; if accepted, production will need this reference to link the reader to the Table.

7. Please include your tables as part of your main manuscript and remove the individual files. Please note that supplementary tables (should remain/ be uploaded) as separate "supporting information" files

8. Please upload a copy of Supporting Information Table S1-S3 which you refer to in your text on page 5.

Reviewers' comments:

Reviewer's Responses to Questions

**Comments to the Author**

1. Is the manuscript technically sound, and do the data support the conclusions?

Reviewer #1: Partly

Reviewer #2: Partly

2. Has the statistical analysis been performed appropriately and rigorously? 

Reviewer #1: N/A

Reviewer #2: N/A

3. Have the authors made all data underlying the findings in their manuscript fully available?

Reviewer #1: No

Reviewer #2: No

4. Is the manuscript presented in an intelligible fashion and written in standard English?

Reviewer #1: No

Reviewer #2: No

5. Review Comments to the Author

Reviewer #1: I have genuinely appreciated the spirit of the paper and the original data presented by the authors of this manuscript.

The topic of B. quintana infection is interesting and the genetic data derived from ancient samples can help science reconstruct the evolutionary history of this pathogen, its interaction with the human species and its global impact on humankind's health.

However, there are some issues that the authors should carefully consider and work on:

1. the English of the manuscript is generally week, with several expressions and collocations typically seen in other languages. In addition, the names of pathogens are not consistently written in italics throughout the manuscript. The manuscript should be read by a native speaker;

2- the tone of the manuscript is often too assertive e.g. "the results reported here are authentic";

3- the general clinical aspects of B. quintana infection should also be presented in the "Introduction" section, as much as the fact that presence of the pathogen does not implicate a severe clinical manifestation. Furthermore, the historical part on the first description of trench fever should be discussed more at length, with a focus on the overall impact on soldiers' health, as much as its interplay with the subsequent Spanish flu pandemic (they are correct in touching co-morbidities later on in the manuscript);

4- talking of comorbidities, the authors should endeavour to speculate more in depth on the potential health consequences on the analyzed individuals;

5- the comparison between infection in soldiers and civilians is interesting but mostly "artificial" because this disease was considered a "military disease" at the very beginning of its recorded history only due to the (special) circumstances under which it was originally described, but there is no real reason to think that poor hygiene conditions are an exclusive characteristic of military life. Way more interesting would be to examine the demographic information for each site, e.g. male/female ratio, distribution according to age range, living standards, diet, detected co-morbidities;

6- in the article, the authors also touch upon the topic of animal infection and zoonotic transmission to the human species, which is indeed a very good point. Further details should be given for the ancient populations analyzed in this study;

7- with reference to the St. Lucia Catacomb (Syracuse, Sicily), from the scientific perspective and that of science communication, I find it controversial that on 8th April 2019 an article (https://www.lasicilia.it/gallery/sicilians/234328/davide-tanasi-l-archeologo-che-dagli-stati-uniti-svela-la-sicilia-antica.html) appeared in the Italian-language press where directly quoted statements were made about the presence in two individuals of "febbre del legionario" - possibly referred to "Legionnaires' disease"?, a completely different disease caused by Legionella pneumophila - while in this manuscript it is written that only 1/28 citizens tested positive.

Are we talking about the same sample(s) presented in this paper or something published elsewhere/unpublished?

If we are talking about the same sample(s), has something changed in the lab data in the meantime? Interestingly, in that press report, a correlation was made between poor diet and disease (exactly what I mentioned in point #5). Why has this not been investigated further in the official scientific publication?

Reviewer #2: It is an interesting manuscript which shows, however, some relevant lacunae.

A. In the abstract the last sentence is not clear: are the Author/s talking about their findings or findings in this area in general? Please clarify this point.

B. Insufficient demographic data are provided for each site in the supplementary material (6 out of 9 sites have no anthropological data). I also suggest to use this data in the context of a lengthier discussion in the main manuscript.

C. The manuscript lacks information about the skeletons and teeth that were examined: for example table 1 should also contain more details about the samples, not only how many and whether they tested positive. In order to better assess the quality of the analyzed sample, I would like to know which specific tooth, from which side, if it is monoradicular or pluriradicular and also the sex and age at death of the skeletons considered. Or you can provide another table with this information. This nice genetic result, to achieve a higher value, should be better related to a proper and exhaustive demographic or, at least when not possible, anthropological discussion.

An addition like this would increase the quality of the manuscript.

D. Do the Author/s have any data on the dietary conditions of these ancient individuals? Literary or archaeological or isotope analyses.

E. In the discussion (page 5), the Author/s write "free of traumatic lesions": can the Author/s better clarify this point? Are the Author/s talking about the absence of pathologies like caries or calculus and also tooth fractures?

F. The English of the manuscript shows some impurities: a check by a native speaker would be a good option.

6. PLOS authors have the option to publish the peer review history of their article (what does this mean?). If published, this will include your full peer review and any attached files.

Reviewer #1: No

Reviewer #2: No

---

## [Author Response · Author response to Decision Letter 0]

24 Jun 2020

Dear Editor,

Please find enclosed the revised version of our manuscript PONE-D-20-07460 entitled “Five millennia of Bartonella quintana bacteremia” along with the answers of the authors to the reviewers’remarks.

Please note that all the authors agreed on the addition of David Peressinotto, and Coralie Demangeot as additional co-authors which now contribution helped to answer some reviewers’ comments.

● A rebuttal letter that responds to each point raised by the academic editor and reviewer(s). This letter should be uploaded as separate file and labeled 'Response to Reviewers'.

● A marked-up copy of your manuscript that highlights changes made to the original version. This file should be uploaded as separate file and labeled 'Revised Manuscript with Track Changes'.

● An unmarked version of your revised paper without tracked changes. This file should be uploaded as separate file and labeled 'Manuscript'.

2. Thank you for including the following ethics statement on the submission details page:

'all the teeth studied, we were provided by their scientific managers having the full

liberty to conducted microbiology studies. This kind of sample is not restricted by any

regulation in force and only requires the authorization of the scientific manager (all

granted as co-authors)'

Please also include this information in the ethics statement in the Methods section of your manuscript. 

Authors’answer: we included this information (lines 75-78)

3. Please include your tables as part of your main manuscript and remove the individual files. Please note that supplementary tables (should remain/ be uploaded) as separate "supporting information" files

"This work was supported by the French Government under the Investissements d’avenir (Investments for the Future) program managed by the Agence Nationale de la Recherche (ANR, fr: National Agency for Research), [reference: Méditerranée Infection 10-IAHU-03]. This work was supported by Région Le Sud (Provence Alpes Côte d’Azur) and European funding [FEDER BIOTK]."

Authors’answer: The text has been corrected accordingly and the appropriate section has been completed in the website.

6. We note you have included a table to which you do not refer in the text of your manuscript. Please ensure that you refer to Table 1-3 in your text; if accepted, production will need this reference to link the reader to the Table.

Authors’answer: The authors corrected mislabelling of Tables, all the three Tables are now cited in the text (Lines 121-125).

7. Please include your tables as part of your main manuscript and remove the individual files. Please note that supplementary tables (should remain/ be uploaded) as separate "supporting information" files

 Authors’answer: Corrected accordingly

8. Please upload a copy of Supporting Information Table S1-S3 which you refer to in your text on page 5.

Authors’answer: The authors corrected mislabelling of Tables, there are NO supplementary Tables. 

 Authors’answer: Corrected accordingly, line 78 in the text (S1 Text) and Supporting Information, line 323-324

Reviewer's Responses to Questions

Comments to the Author

Reviewer #1: I have genuinely appreciated the spirit of the paper and the original data presented by the authors of this manuscript.

The topic of B. quintana infection is interesting and the genetic data derived from ancient samples can help science reconstruct the evolutionary history of this pathogen, its interaction with the human species and its global impact on humankind's health.

However, there are some issues that the authors should carefully consider and work on:

1. the English of the manuscript is generally week, with several expressions and collocations typically seen in other languages. In addition, the names of pathogens are not consistently written in italics throughout the manuscript. The manuscript should be read by a native speaker;

Authors’answer: The revised version of the manuscript has been corrected by professionals (Certificate enclosed).

2- the tone of the manuscript is often too assertive e.g. "the results reported here are authentic";

Authors’answer: The revised version of the manuscript has been corrected by professionals who took this appreciation (which is just incomprehensible for French authors) into consideration (Certificate enclosed).

3- the general clinical aspects of B. quintana infection should also be presented in the "Introduction" section, as much as the fact that presence of the pathogen does not implicate a severe clinical manifestation. Furthermore, the historical part on the first description of trench fever should be discussed more at length, with a focus on the overall impact on soldiers' health, as much as its interplay with the subsequent Spanish flu pandemic (they are correct in touching co-morbidities later on in the manuscript);

Authors’answer:

- The clinical manifestations of infection presented in the Introduction section are now expanded as proposed by the reviewer (Lines 55-58). 

- The authors do not agree with the following remark of the reviewer that B. quintana infection is a mild one. The authors are now reporting on mortality rates which clearly indicate that B. quintana infection is indeed a life-threatening infection (Lines 52-55).

- The authors carefully checked that no report issued regarding B. quintana and Spanish flu (see reference [2,3], line 211-213).

4- talking of comorbidities, the authors should endeavour to speculate more in depth on the potential health consequences on the analyzed individuals;

Authors’answer: The authors thanks the reviewer to give the opportunity to discuss this interesting aspect (Lines 139-142).

5- the comparison between infection in soldiers and civilians is interesting but mostly "artificial" because this disease was considered a "military disease" at the very beginning of its recorded history only due to the (special) circumstances under which it was originally described, but there is no real reason to think that poor hygiene conditions are an exclusive characteristic of military life. Way more interesting would be to examine the demographic information for each site, e.g. male/female ratio, distribution according to age range, living standards, diet, detected co-morbidities;

Authors’answer: Indeed, one output of the present study was to demonstrate an absence of significant difference between civil and military populations; hence suggesting the absence of differences in terms of cloth hygiene (Lines 180-182).

6- in the article, the authors also touch upon the topic of animal infection and zoonotic transmission to the human species, which is indeed a very good point. Further details should be given for the ancient populations analyzed in this study;

Authors’answer: The authors just not have any data regarding contacts with animals in any of the buried populations here reported.

7- with reference to the St. Lucia Catacomb (Syracuse, Sicily), from the scientific perspective and that of science communication, I find it controversial that on 8th April 2019 an article (https://www.lasicilia.it/gallery/sicilians/234328/davide-tanasi-l-archeologo-che-dagli-stati-uniti-svela-la-sicilia-antica.html) appeared in the Italian-language press where directly quoted statements were made about the presence in two individuals of "febbre del legionario" - possibly referred to "Legionnaires' disease"?, a completely different disease caused by Legionella pneumophila - while in this manuscript it is written that only 1/28 citizens tested positive.

Are we talking about the same sample(s) presented in this paper or something published elsewhere/unpublished?

If we are talking about the same sample(s), has something changed in the lab data in the meantime? Interestingly, in that press report, a correlation was made between poor diet and disease (exactly what I mentioned in point #5). Why has this not been investigated further in the official scientific publication?

Authors’answer: 

- The reviewer is perfectly right to point to the possibility of co-infection with Bartonella quintana in ancient populations, and this interesting point is now emphasized in the revised version of the manuscript (Lines 139-142).

- Then, to the best of the authors knowledge, co-infections with Bartonella quintana are with other lice-borne pathogens including Yersinia pestis, Rickettsia prowazekii; this point is now clarified in the revised version of the manuscript (Lines 139-142).

- Then, to the best of the authors knowledge, Legionella pneumophila is NOT known to be a lice-borne pathogen; but rather a water-borne pathogen currently transmitted by aerosols of contaminated water.

- Therefore, there was no particular reason to test for the hypothesis of Legionella pneumophila rather than any other hypothesis.

- At last, the “La Sicilia” paper referred to by the reviewer, is clearly grey literature which cannot be incorporated nor discussed as a reference in the present work to be published in PLoS ONE.

- The authors did not test for Legionella pneumophila but rather for Bartonella quintana; the 1/28 individual in Table 1 is referring to Bartonella quintana, not at all to Legionella pneumophila. 

Reviewer #2: It is an interesting manuscript which shows, however, some relevant lacunae.

A. In the abstract the last sentence is not clear: are the Author/s talking about their findings or findings in this area in general? Please clarify this point.

Authors’answer: The reviewer is perfectly right and this sentence was deleted from the revised version.

B. Insufficient demographic data are provided for each site in the supplementary material (6 out of 9 sites have no anthropological data). I also suggest to use this data in the context of a lengthier discussion in the main manuscript.

Authors’answer: The reviewer is right but no further demographic data were available in the reasonable time for revising this manuscript. 

C. The manuscript lacks information about the skeletons and teeth that were examined: for example table 1 should also contain more details about the samples, not only how many and whether they tested positive. In order to better assess the quality of the analyzed sample, I would like to know which specific tooth, from which side, if it is monoradicular or pluriradicular and also the sex and age at death of the skeletons considered. Or you can provide another table with this information. This nice genetic result, to achieve a higher value, should be better related to a proper and exhaustive demographic or, at least when not possible, anthropological discussion.

An addition like this would increase the quality of the manuscript.

Authors’answer: The reviewer is right but no further demographic data were available in the reasonable time for revising this manuscript. 

D. Do the Author/s have any data on the dietary conditions of these ancient individuals? Literary or archaeological or isotope analyses.

 Authors’answer: The reviewer is right but no further demographic data were available in the reasonable time for revising this manuscript. 

E. In the discussion (page 5), the Author/s write "free of traumatic lesions": can the Author/s better clarify this point? Are the Author/s talking about the absence of pathologies like caries or calculus and also tooth fractures?

Authors’answer: The reviewer is perfectly right, this point is clarified both in the Discussion section (Lines 128-130).

F. The English of the manuscript shows some impurities: a check by a native speaker would be a good option.

Authors’answer: The revised version of the manuscript has been corrected by professionals (Certificate enclosed).

As the authors answered all the Editor and Reviewers’ remarks and corrected the manuscript accordingly, they hope that this revised version will be accepted for publication in PLoS ONE.

Sincerely,

Dr. Gérard ABOUDHARAM,

Corresponding author.

---

## [Decision Letter · Decision Letter 1]

11 Aug 2020

PONE-D-20-07460R1

Five millennia of Bartonella quintana bacteremia.

PLOS ONE

Dear Dr. Aboudharam,

Thank you for submitting your manuscript to PLOS ONE. After careful consideration, we feel that it has merit but does not fully meet PLOS ONE’s publication criteria as it currently stands. Therefore, we invite you to submit a revised version of the manuscript that addresses the minor point highlighted by one referee that are reported below .

We look forward to receiving your revised manuscript.

Kind regards,

David Caramelli, Ph.D

Academic Editor

PLOS ONE

Reviewers' comments:

Reviewer's Responses to Questions

**Comments to the Author**

1. If the authors have adequately addressed your comments raised in a previous round of review and you feel that this manuscript is now acceptable for publication, you may indicate that here to bypass the “Comments to the Author” section, enter your conflict of interest statement in the “Confidential to Editor” section, and submit your "Accept" recommendation.

Reviewer #1: All comments have been addressed

Reviewer #2: (No Response)

2. Is the manuscript technically sound, and do the data support the conclusions?

Reviewer #1: Yes

Reviewer #2: Yes

3. Has the statistical analysis been performed appropriately and rigorously? 

Reviewer #1: N/A

Reviewer #2: N/A

4. Have the authors made all data underlying the findings in their manuscript fully available?

Reviewer #1: Yes

Reviewer #2: Yes

5. Is the manuscript presented in an intelligible fashion and written in standard English?

Reviewer #1: Yes

Reviewer #2: Yes

6. Review Comments to the Author

Reviewer #1: This new version of the manuscript is well-researched and presented in a better form. In my opinion, the authors have made good amendments.

The "La Sicilia" reference be may grey literature but I am very skeptical about going to the media with certain claims and then issuing something different in a scientific paper, but thanks for clarifying that.

I would only recommend adding a couple of general references when you discuss the pathogen-population interaction:

1) Bos KI, Kühnert D, Herbig A, et al. Paleomicrobiology: Diagnosis and Evolution of Ancient Pathogens. Annu Rev Microbiol. 2019;73:639-666.

2) Rühli FJ, Galassi FM, Haeusler M. Palaeopathology: Current challenges and medical impact. Clin Anat. 2016;29(7):816-822.

Reviewer #2: Edits made in an orderly and clear way. I think the authors have answered the main critical points of the previous version of this manuscript.

7. PLOS authors have the option to publish the peer review history of their article (what does this mean?). If published, this will include your full peer review and any attached files.

Reviewer #1: No

Reviewer #2: No

---

## [Author Response · Author response to Decision Letter 1]

18 Aug 2020

Dear Editor,

Please find enclosed the second revised version of the manuscript PONE-D-20-07460 entitled: “Five millennia of Bartonella quintana bacteremia” along with the answers of the authors to the reviewers ‘ comments.

Reviewers' comments:

Reviewer #1: This new version of the manuscript is well-researched and presented in a better form. In my opinion, the authors have made good amendments.

The "La Sicilia" reference be may grey literature but I am very skeptical about going to the media with certain claims and then issuing something different in a scientific paper, but thanks for clarifying that.

Authors’answer: The authors acknowledge this positive remark.

I would only recommend adding a couple of general references when you discuss the pathogen-population interaction:

1) Bos KI, Kühnert D, Herbig A, et al. Paleomicrobiology: Diagnosis and Evolution of Ancient Pathogens. Annu Rev Microbiol. 2019;73:639-666.

2) Rühli FJ, Galassi FM, Haeusler M. Palaeopathology: Current challenges and medical impact. Clin Anat. 2016;29(7):816-822.

Authors’answer: The reviewer is perfectly suggesting to broaden the Discussion section, these two references are now cited in the Discussion section (Lines 213-215), and listed as new references 44 and 45, accordingly.

Reviewer #2: Edits made in an orderly and clear way. I think the authors have answered the main critical points of the previous version of this manuscript.

Authors’answer: The authors acknowledge this positive remark.

As the authors answered the Reviewer n°1 remark and corrected the manuscript accordingly, the authors hope that this second revised version will be accepted for publication,

Sincerely

Gérard ABOUDHARAM,

Corresponding author.

---

## [Editor Report · Decision Letter 2]

9 Sep 2020

Five millennia of Bartonella quintana bacteremia.

PONE-D-20-07460R2

Dear Dr. Aboudharam,

We’re pleased to inform you that your manuscript has been judged scientifically suitable for publication and will be formally accepted for publication once it meets all outstanding technical requirements.

Kind regards,

David Caramelli, Ph.D

Academic Editor

PLOS ONE
---

## [Editor Report · Acceptance letter]

7 Oct 2020

PONE-D-20-07460R2 

Five millennia of *Bartonella quintana* bacteraemia. 

Dear Dr. Aboudharam:

I'm pleased to inform you that your manuscript has been deemed suitable for publication in PLOS ONE. Congratulations! Your manuscript is now with our production department. 

Kind regards, 

on behalf of

Professor David Caramelli 

Academic Editor

PLOS ONE